# Recommended Values for the Hydrophobicity and Mechanical Properties of Coating Materials Usable for Preparing Controlled-Release Fertilizers

**DOI:** 10.3390/polym15244687

**Published:** 2023-12-12

**Authors:** Yajing Wang, Juan Li, Ru Lin, Dianrun Gu, Yuanfang Zhou, Han Li, Xiangdong Yang

**Affiliations:** State Key Laboratory of Efficient Utilization of Arid and Semi-arid Arable Land in Northern China/Key Laboratory of Plant Nutrition and Fertilizer, Ministry of Agriculture and Rural Affairs/the Institute of Agricultural Resources and Regional Planning, Chinese Academy of Agricultural Sciences, Beijing 100081, China; 18837129765@163.com (Y.W.); lijuan02@caas.cn (J.L.); mumuru666@163.com (R.L.); dianrunna@163.com (D.G.); zyf20220129@163.com (Y.Z.); qhxna2288@163.com (H.L.)

**Keywords:** controlled-release fertilizer, coating material, hydrophobicity, mechanical property

## Abstract

The hydrophobicity and mechanical properties of coating materials and the nitrogen (N) release rates of 11 kinds of controlled-release fertilizers (CRFs) were determined in this study. The results show that the N release periods of the CRFs had negative correlations with the water absorption (WA) of the coating materials (y = 166.06x^−1.24^, r = 0.986), while they were positively correlated with the water contact angle (WCA) and elongation at break (EB) (y = 37.28x^0.18^, r = 0.701; y = −19.42 + 2.57x, r = 0.737). According to the fitted functional equation, CRFs that could fulfil the N release period of 30 days had a coating material WA < 2.4%, WCA > 68.8°, and EB > 57.7%. The recommended values for a CRF that can fulfil the N release period of 30 days are WA < 3.0%, WCA > 60.0°, and EB > 30.0% in the coating materials. CRFs with different nutrient release periods can be designed according to the recommended values to meet the needs of different crops. Furthermore, our experiments have illustrated that the N release period target of 30 days can be reached for modified sulfur-coated fertilizers (MSCFs) by improving their mechanical properties.

## 1. Introduction

Food security strongly depends on a sufficient nutrient supply [1,2]. However, the excess use of commercial fertilizers for maximum agricultural crop production leads to serious resource consumption and severe environmental pollution [3,4], as well as directly or indirectly affecting human health [5]. A one-off application of a controlled-release fertilizer (CRF) can meet the nutrient requirements of crops [6]. It has been illustrated that the use of CRF is a green technology that not only improves fertilizer use efficiency, reduces nitrogen (N) loss, and saves labor [7], but also unifies these factors with its environmental benefits and economic benefits [8]. The development of CRFs mainly depends on the development process of coating materials, ranging from inorganic to organic and from natural polymers to organic synthetic polymers [9]. To date, only a few polymers, such as alkyd resin [10,11], polyethylene (PE) [12], polyurethane (PU) [13], and styrene–acrylic latex (SAL) [14], have exhibited excellent controlled-release effects. Varieties of inorganic, organic, and biological materials, such as sulfur (S), calcium magnesium phosphate rock powder, polyvinyl alcohol (PVA), natural cellulose, lignin, chitin, and tung oil, have various drawbacks [15]. For example, the film structure collapses when in contact with water, swelling or rupturing when absorbing the water due to its poor ability to hinder water and fertilizer; in turn, it exhibits poor controlled-release characteristics.

The nutrient release of CRFs has been found to be closely associated with their film structure [16,17,18] and the characteristics of the coating materials [6]. Previous studies have assumed that the film structure remains stable when a CRF is immersed in water under ideal conditions, and several models have been developed to explain the mechanism of nutrient diffusion based on this assumption [16,19,20]. However, establishing a comprehensive release mechanism is challenging due to uncertain external environmental factors and material properties. In reality, most film properties undergo changes upon contact with water [21], resulting in alterations in both film structure and nutrient release rate. Therefore, further improvements are needed for existing release models [22,23,24], particularly considering the complex release process of swelling film materials [25]. Consequently, it can be concluded that the performance of controlled-release coatings primarily depends on their material properties.

In the past, it was widely believed that the hydrophobicity [26,27] and mechanical properties [28] of coating materials were crucial factors in controlling nutrient release. For instance, the water absorption (WA) of a starch/PVA-CRF coating film exceeded 360% [29], while PVA/biochar composite coating materials exhibited WA values ranging from 200 to 300% [30]. Similarly, biodegradable starch/PVA/bentonite graft polymers displayed WA values within the range of 200–300% [31]. Furthermore, double-coated slow-release fertilizers developed with ethyl cellulose ether and starch-based highly absorbent polymers demonstrated approximately 100% WA [9]. These films had high hygroscopicity and rapidly absorbed water, resulting in reduced controllability. Therefore, efforts were made to enhance their controllability by reducing their WA. For example, a composite film modified with nano-silica and γ-polyglutamic acid achieved substantial reductions in its WA, along with enhanced densification compared to unmodified PVA film; this modification led to improved controllability [32]. Additionally, etherified epoxy resin was employed in modifying the PVA film, which resulted in lower WA (884 g/kg) than unmodified PVA at the same concentration; furthermore, a CRF coated with modified PVA exhibited a slow nutrient release performance [33].

The water contact angle (WCA) serves as a crucial indicator of the hydrophobic performance of materials, with larger WCA values indicating stronger water barrier properties [34]. However, bio-based materials exhibit smaller WCAs and shorter nutrient release periods. To optimize their water resistance, researchers have enhanced the characteristics of their coating materials. By modifying the hydrophobic properties of these coatings, it is possible to significantly increase the WCA and greatly improve the controlled-release performance. For instance, Xie et al. developed a biomimetic bio-based CRF by incorporating a superhydrophobic film surface modified with micro/nanoscale diatomaceous earth silica. This modification reduced the nutrient release rate and extended the nutrient release period twofold compared to an unmodified CRF [35]. The surface energy of the bio-based CRF was reduced and the WCA increased from 89.8 to 158.9° after modification, resulting in an extended nutrient release period of the modified CRF with a 3% coating rate from 5 days to approximately 28 days [36]. Additionally, the hydrophobicity of PU was enhanced through the use of paraffin wax as well as nano-silica fillers or by incorporating hydroxypropyl-capped polydimethylsiloxane (HP-PDMS) into the PU reaction [37,38,39]. Furthermore, when nano-cellulose crystals were added to PVA with high WA, the WCA of PVA increased from 29.3 to 69.7° and led to an approximately twofold extension in the nutrient release period for the modified CRF [40]. These findings demonstrate that hydrophobic modifications in coating materials can significantly enhance the nutrient release performance of CRFs.

Usually, materials with simultaneous good hydrophobicity and poor mechanical properties often lack controllability. In most coating processes, nanomaterials or other modifiers are employed to enhance the mechanical properties of coatings. For instance, when S was utilized as a coating to derive sulfur-coated fertilizers (SCFs) with a WCA of 78°, it exhibited favorable water resistance; however, the SCFs had inadequate mechanical properties. By modifying S with dicyclopentadiene (DCPD), the compressive strength increased from 27 to 47 N, and the 7-day cumulative release rate of N from the SCFs decreased from 83 to 54% [41]. Consequently, the mechanical modification of S to yield good hydrophobic properties resulted in an improvement in its controlled-release performance. Furthermore, for polymeric materials, enhancing their mechanical properties can also lead to improved controlled-release characteristics. For example, silica-modified PU demonstrated an increase in tensile strength (TS) from 40 to 54 MPa while decreasing its elongation at break (EB) from 7 to 2%, thereby extending the nutrient release period by approximately 32% [42]. Moreover, compared to the pure PU material, incorporating modified natural pyrophyllite into a PU composite enhanced both its TS and EB by approximately 0.61 MPa and 42%, respectively, and increased the nutrient release period of the CRF by an additional 15 days [43].

The unmodified materials mentioned above all exhibited drawbacks in terms of hydrophobicity and mechanical properties, resulting in poor controllability and the need for improvements in their nutrient release periods through targeted modifications. However, it was found that unmodified PE with enhanced hydrophobicity and mechanical properties could achieve an N release period of eight months [17]. Therefore, achieving the good controllability of coating materials requires ensuring, at least to some degree, their hydrophobic and mechanical characteristics simultaneously. In other words, there exists an intrinsic relationship between the hydrophobicity and mechanical properties of the coating materials and the nutrient release period of the CRF. However, it remains unclear to what extent the hydrophobicity and mechanical properties of the coating materials need to be optimized to ensure good controllability. Thus, we hypothesized that (1) a correlation can be established between the hydrophobicity and mechanical properties of coating materials and the nutrient release period of CRFs; (2) recommended values for an excellent controlled-release performance can be determined for both hydrophobicity and the mechanical properties; (3) these recommended values can be experimentally validated for reliability. The objective of this study was to determine the values of both the hydrophobicity and mechanical properties that are necessary for preparing CRFs using suitable coating materials. Firstly, we measured the films’ hydrophobicity, mechanical properties, and N release periods based on existing CRF products. Subsequently, we established a numerical fitting method to establish relationships between these parameters. Secondly, we calculated the parameter values corresponding to N release periods of 30 or 90 days, respectively. Finally, the reliability of the relational model was validated through enhancements in the physical properties of the materials, ensuring their adherence to recommended values. This relational model can effectively determine the mechanical properties and hydrophobic value required for achieving an exceptional release performance in CRFs, thereby offering valuable parameter guidance for coating material development.

## 2. Materials and Methods

### 2.1. Experimental Materials, Equipment, and Devices

Urea granules (2–4 mm diameter, 46% nitrogen content, Shandong Hualu-Hengsheng Group Co., Ltd., Dezhou, China); polypropylene glycol (PPG, molecular weight (MW) = 400, Sinopec Asset Management Co., Ltd., Tianjin Petrochemical Branch, Tianjin, China); 4,4′-diphenylmethane diisocyanate (MDI, Wanhua Chemical Group Co., Ltd., Yantai, China); polytetrahydrofuran (PTMG, MW = 250, Xuzhou Yihui Yang New Material Co., Ltd., Xuzhou, China); polyethylene glycol (PEG, MW = 200, Lotte Chemical, Seoul, Republic of Korea); polycapro-lactone (PCL1,MW = 500; PCL2, MW = 1000, Hunan Juren Chemical New Material Technology Co., Ltd., Changsha, China); poly(1,4′-butylene adipate) (PBA, MW = 1000, Shandong Jiaying Chemical Co., Ltd., Qingdao, China); polyvinyl alcohol (PVA, MW = 1750, Shanghai Maclean Biochemical Technology Co., Ltd., Shanghai, China); polyethylene (PE, Sinopec Beijing Yanshan Petrochemical Co., Beijing, China); styrene–acrylic latex 1 (SAL1, PRIMAL AS-2010, Dow Chemical Co., Shanghai, China); styrene–acrylic latex 2 (SAL2, PRIMAL AS-8098, Dow Chemical Co., Shanghai, China); sulfur (S, Beijing Jinyuanteng Trading Co., Ltd., Beijing, China); sulfur-coated fertilizer (SCF, Shanghai Hanfeng Slow Release Fertilizer Co., Ltd., Shanghai, China); 4-dimethylaminobenzaldehyde (PDAB, AR, Shanghai Maclean Biochemical Technology Co., Ltd., Beijing, China); nano-silica (SiO_2_, Jiangsu Xianfeng Nanomaterials Technology Co., Ltd., Nanjing, China).

Microcomputer heating platform (GVECTECH V3030T, Dingxinyi Experimental Equipment Co., Ltd., Shanghai, China); air-drying oven (Shanghai Lichen Instrument Technology Co., Ltd., Shanghai, China); ultraviolet spectrophotometer (SPECORD200, Analytik Jena, Jena, Germany); electronic balance (ME3002, Mettler Toled, Germany); spiral micrometer (0–25 mm, 0.001 mm, Nanjing Sutech Measuring Instruments Co., Ltd., Nanjing, China); contact angle measuring instrument (SCA20, Data Physics, Filderstadt, Germany); vertical computer servo material testing machine (CREE-8003A, Dongguan Krui Instrument Co., Ltd., Guangzhou, China); small fluidized bed (New Fertilizer Experimental Base of International Agricultural High-tech Industrial Park, Chinese Academy of Agricultural Sciences, self-made, Bejing, China); coating pan (YB400, Henan Qineng Machinery and Equipment Co., Ltd., Zhengzhou, China).

### 2.2. Experimental Methods

#### 2.2.1. Preparation of Films

Preparation of PVA film: A PVA solution was prepared by completely dissolving 2.95 g of PVA granules in 46 mL of deionized water at 100 °C, followed by the addition of 1.05 g of polyvinyl pyrrolidone under stirring at 90 °C for 2 h. Subsequently, the resulting PVA coating solution was poured onto a Teflon board (dimensions: 25 cm × 25 cm × 0.1 mm) and allowed to flow naturally. The films were then placed on a microcomputer heating platform at 50 °C for a reaction time of 2 h. Finally, the PVA films were obtained after drying them in an air-drying oven at 50 °C for a duration of 12 h (Figure 1).

Preparation of SAL1 film: A 30% dry matter content SAL1 solution was prepared by adding 66.7 mL of water to 100 g of SAL1 with a 50% dry matter content. After thorough mixing, the mixture was poured onto a Teflon board (25 cm × 25 cm × 0.1 cm) and allowed to flow naturally before being placed on a micro-computer heating platform at 50 °C for 2 hours. The SAL1 film was obtained after drying the films in an air-drying oven at 50 °C for 12 hours (Figure 1).

Preparation of PE film: A PE solution with a concentration of 8% (*w*/*w*) was prepared by dissolving PE granules into a heated tetrachloroethylene solution at 80–90 °C, resulting in a homogeneous mixture. The mixture was then stirred and raised to 120 °C, maintaining this temperature for 20–30 min to ensure complete dissolution of the PE granules. Subsequently, the resulting PE coating solution was poured onto a Teflon board, allowing for the formation of a PE film through the complete volatilization of tetrachloroethylene.

Preparation of PU films: The PU films were prepared using a one-step synthetic method. Polyol (PPG, PTMG, PBA, PEG, and PCL) and MDI were weighed according to a molar ratio of 1:1 (Table 1) and thoroughly mixed. The mixture was then poured onto a Teflon board (dimensions: length × width × depth = 25 cm × 25 cm × 0.1 cm) to allow natural flow. Subsequently, the films were placed on a micro-computer heating platform at 50 °C for 2 h to initiate the reaction. Following drying at 50 °C for an additional 12 h in an air-drying oven, six types of PU films (PPG, PTMG, PBA, PEG, PCL1, and PCL2) were obtained based on the formula for PU coating materials.

#### 2.2.2. Preparation of CRFs in the Fluidized Bed Spray-Coating Process

The Wurster fluidized bed was employed as the coating apparatus (Figure 1), and diverse materials were utilized to configure the coating solution in accordance with the methodology outlined in Section 2.2.1, while a comprehensive scheme detailing material proportions can be found in Table 1.

Preparation of PVA-coated CRF (PVACF): The process flow chart for preparing PVACF is presented in Figure 1. Initially, the temperature of the coated fluidized bed was preheated to 80 °C, and large urea granules were added based on the formula design provided in Table 1 until a steady state of fluidization was achieved. Subsequently, liquid paraffin was introduced into the fluidized bed and mixed with the urea particles for 2 min. PVACF was then prepared by adding a PVA solution with a concentration of 5%, which was controlled using a peristaltic pump and pumped into the two-fluid nozzle at the bottom of the fluidized bed at a rate of 30 mL/min. The solution was atomized and sprayed onto the surface of the urea particles to coat them, resulting in the CRF’s preparation. 

Preparation of SAL-coated CRF (SALCF): The process flow chart for the preparation of SALCF is illustrated in Figure 1. The materials were added based on the formula design provided in Table 1, and both SALCF1 and SALCF2 were prepared following the same procedure as PVACF. 

Preparation of PE-coated CRF (PECF): The process flow chart of preparing the PECF is shown in Figure 1. The materials were added according to the formula design in Table 1 and the PECF was prepared in the same way as PVACF [44].

Preparation of PU-coated CRFs (PUCFs): The PUCFs were prepared using the in situ reaction method [37]. The process flow chart for preparing the PUCFs is illustrated in Figure 1. Initially, the coated fluidized bed was preheated to 70 °C, and large urea granules were added according to the formula design specified in Table 1 until a steady state of fluidization was achieved. Subsequently, liquid paraffin was introduced into the fluidized bed and mixed with the urea granules for a duration of 2 min. The PUCFs were then synthesized through a dip-coating process involving polyol (PPG, PTMG, PBA, PEG, PCL1, and PCL2) and MDI addition. The reaction proceeded within the fluidized bed for approximately 20 min before removing the resulting PUCFs. This procedure yielded six distinct types of PUCFs.

#### 2.2.3. Preparation of Modified S Film and Modified SCFs (MSCF) 

Preparation of modified S film: S powder and PCL1 were accurately weighed to the weight ratios of MSCF1—S:PCL1:MDI = 15:9.91:5.09 and MSCF2—S:PCL1:MDI = 15:4.96:2.54 and thoroughly mixed to achieve homogeneity. Subsequently, MDI was added to the mixture to formulate the coating materials, which were then applied onto a flat film using a steel wire rod coater. A round pot coating machine containing 500 g of large-particle urea was preheated at 75 °C for 2 min. The atomizing nozzle was employed to spray the blend of PCL1 and S powder onto the surface of the large-particle urea while simultaneously introducing MDI into the coating pan for the urea coating operation.

Preparation of MSCF: The round pot coating machine was preheated at 75 °C for 2 min with the addition of 500 g of large-particle urea. A mixture of S powder and PCL1 was sprayed onto the surface of the large-particle urea using an atomizing nozzle, while MDI was introduced into the coating pan for the urea coating operation.

#### 2.2.4. Determination of the N Release Period of the CRFs 

The N release period of the CRFs was determined using the water immersion method at 25 °C. The content of urea–N in water was spectrophotometrically determined in accordance with regulation ISO-18644-2016 [45]. The nutrient release index formula is as follows:(1)$$\eta_{t1}=\frac{m_{t1}}{M}\times 100\%$$
(2)$$\eta =\frac{\sum \eta_{m}}{M}\times 100\%$$
(3)$$\eta_{∆t}=\frac{\eta_{tn}-\eta_{t1}}{t_{n}-t_{1}}\times 100\%$$
(4)$$T=1+\frac{80\%-\eta_{t1}}{\eta_{∆t}}\;$$

*η_t_* is the N cumulative release rate; *t*_1_ is the first day; *t_n_* is n days after incubation; *η_t_*_1_ is the initial N release rate; *η_tn_* is the N cumulative release rate on the nth day; *m_t_*_1_ is the N content released on the 1st day; *η_m_* is the N content released over n days; *M* is the total N content released in the CRFs; *η_*∆*t_* is the average N release rate; and *T* is the N release period.

#### 2.2.5. Measurement of the WA of Films

The WA of films could be evaluated using the gravimetric method according to GB/T 1034-2008. Dried samples were weighed in triplicate, then immersed in 200 mL distilled water at 23 ± 1 °C for 24 h. The samples, taken out using tweezers, were weighed after filtering, and the WA of the films was calculated based on Equation (5):(5)$$WA=\frac{M_{2}-M_{1}}{M_{1}}\times 100\%$$
where *M*_1_ and *M*_2_ refer to the dry and wet weights of the films, respectively.

#### 2.2.6. Measurement of the WCA of Films

The contact angle, ranging from 0 to 180°, represents the angle formed at the interface of the solid, gas, and liquid phases. It characterizes the hydrophobicity of the controlled-release films by measuring their WCA using an instrument at ambient temperature. The average WCA values were obtained through measuring at five different positions on the same sample with 4 μL water droplets.

#### 2.2.7. Measurement of the Mechanical Properties of Films

The mechanical properties of the films were evaluated using a vertical computer servo material testing machine in accordance with GB/T 1040-2018, with five replicates. The films were shaped like dumbbells and their thickness was measured using a spiral micrometer. The initial gauge length was set at 25 mm, and the measuring speed was maintained at 300 mm/min. EB and TS values were determined based on five independent drawing experiments performed under identical conditions.

#### 2.2.8. Statistical Analyses

The data processing and statistical analyses were conducted using Microsoft Excel 2013 (Microsoft Corporation, Redmond, WA, USA). Figures were generated using Origin2018 (Origin Lab, Northampton, MA, USA). Pearson correlation analysis was employed to examine the respective relationships between the N release periods of CRFs and the WCA, WA, EB, and TS of the coating materials.

## 3. Results

### 3.1. The N Release Characteristics of CRFs

The N release characteristics of 11 types of CRFs and their N cumulative release curves are shown in Table 2 and Figure 2, respectively.

PECF exhibited *η_t_*_1_ and *η_*∆*t_* values of 0.6% and 0.4%, respectively, indicating a N release period of approximately 178 days; PPGCF coated with polypropylene glycol also displayed a slower N release characteristic, with an N cumulative release rate of 41.0% at day 28 and a N release period of around 54 days. Both PECF and PPGCF met the CRF (ISO-18644) release criteria, and their N release curves were nearly linear within the first 28 days, demonstrating good controllability. These findings suggest their widespread use, as primary products currently available on the market, in agriculture.

The two types, SALCF1/SALCF2, exhibited favorable N controlled-release properties, albeit with slightly different N release periods. Figure 2 illustrates the N release curve of SALCF2, which follows an inverse “L” shape characterized by rapid initial N release followed by a gradual decline, with a total N release period of approximately 34 days. On the other hand, SALCF1 demonstrated an N release rate equivalent to 0.2% for *η_t_*_1_ and 1.6% for *η_*∆*t_*, resulting in a cumulative N release rate of 44.1% over a span of 28 days and an overall N release period approaching 50 days, thereby satisfying the nutrient demands of agricultural crops. The N release periods of different PUCFs prepared with various polyols as soft segments exhibited significant variations, as illustrated in Figure 2. PUCF formulated with PPG as a soft segment demonstrated a N release period of 54 days. Conversely, the three types of PUCFs incorporating low-molecular-weight polyols (PTMG/PEG/PCL1) as soft segments displayed comparable N release periods ranging from approximately 26 to 28 days. However, when employing PCL2 and PBA with an MW of 1000 as soft segments, the resulting *η*_1_ values for the respective PUCFs were determined to be 55.1% and 27.8%. Furthermore, during a span of four days, the cumulative N release rates reached an alarming level of 78.2% and 78.5%, respectively, indicating that these two types of PUCFs exhibit uncontrollable behavior.

In addition, the PVACF exhibited a rapid trend of N release with a complete period of approximately 1.5 days for N release. The *η*_1_ value of the SCF was determined to be 17.7%, followed by a significantly slower release rate thereafter. Two distinct leaching mechanisms were observed for the SCF in water, characterized by “all or nothing” performances [46]. Firstly, certain portions of the SCF surface easily crumbled and formed large pores, facilitating the quick dissolution of the fertilizer core and transforming the SCF into an empty shell; this phenomenon accounted for the initially high N release rate observed for the SCF. Secondly, other parts of the SCF remained intact even after one month’s immersion in water, indicating no subsequent release.

### 3.2. The Hydrophobicity of Different Coating Materials

In order to investigate the relationship between the release performance of the CRFs and the properties of the materials, flat films with different formulations were prepared and the WA and WCA of the films were measured (Figure 3a,b). The WA of different flat films varied significantly from 104.0% for PVA to 0.2% for PE. The highest WA of PVA at ambient temperature could be associated with the presence of a large number of hydroxyl groups in the PVA molecular chain. SAL1, which is a water-based material, also had a WA of 9.4%. The WAs of different PU films with different soft sections ranged from 1 to 6%. The PE film was hydrophobic in water, and the WA of the S film was similar to that of the PE film. The WCA of these films ranged from 36.1 to 92.9°. Briefly, the smallest WCA for the PVA film was 36.1° and the smallest for the SAL1 film was 66.5°. The WCAs of PU films with different soft sections were in the range of 53.5 to 83.4°. The PE film showed the highest WCA, and the S film had a similar WCA value to the PE, of 92.1°.

The relationship between the WCA, WA, and hydrophobicity is illustrated in Figure 3. A higher WCA corresponds to a lower WA and stronger hydrophobicity. For instance, the PE film exhibited a large WCA of 92.9° and an extremely low WA of 0.04%, indicating its excellent controlled-release properties. In contrast, the PVA film had a significantly smaller WCA of 36.1° and a remarkably high WA of 104.0%, suggesting poor water resistance and uncontrolled behavior. PU films with different soft sections generally exhibited higher WCAs but lower WAs. The WCA of the PCL1 film was 83.4°; however, its WA was low, at 1.7%, while the PEG film showed a decreased WCA of 53.5° and an elevated WA of 5.5% compared to the PCL1 film. All these results suggest close mutual restraint between the hydrophobicity of the materials and the nutrient release period of CRFs.

### 3.3. Mechanical Properties of Different Coating Materials

The results of the EB and TS for different coating materials are presented in Figure 3c,d, suggesting the significant influence of material type on the mechanical properties of the composite films.

The three coating materials, namely, PCL2, SAL1, and PE, exhibited favorable controlled-release performances, with EB values exceeding 30%. Notably, materials with a TS ranging from 9.7 to 12.2 MPa, such as SAL1 and PE, demonstrated superior controlled-release capabilities. Among these materials, the PE film stood out due to its exceptional mechanical properties, characterized by an impressive EB value of 466.8% and a TS value of 9.7 MPa. 

Additionally, the SAL1 film exhibited a relative reduction in EB (159%) and an increase in TS (12.2 MPa), resulting in a slightly stiffer and more brittle texture compared to the PE film. Consequently, its mechanical properties and controlled-release performance also showed a corresponding decline. However, the PCL2 film with a higher EB showed inadequate strength, resulting in a suboptimal controlled-release performance.

Poor controllability was observed in almost all of the films with an EB <30%, including PVA, PBA, PEG, PCL1, PTMG, and PPG. Notably, the PVA and PBA films, with TS values of 25.6 and 27.4 MPa, respectively, exhibited a hard and brittle nature without any significant control. Conversely, the PEG and PPG films demonstrated excellent controlled-release performances, with TS values of 2 and 3 MPa, respectively. Similarly, the PTMG film displayed good controllability, with a TS value of 48.9 MPa. In conclusion, no definite relationship between the TS and EB could be established; however, materials exhibiting moderate stiffness combined with high flexibility are more suitable for use in coating applications.

### 3.4. Correlations between the Controlled-Release Performance of CRFs and the Physical Properties of the Coating Materials

In order to investigate the relationship between the controlled-release performance of CRFs and the hydrophobic and mechanical properties of the coating materials, we analyzed the correlations between the N release periods of CRFs and the WCA, WA, EB, and TS of the coating materials (Figure 4).

Figure 4 demonstrates that the correlation between the N release periods of CRFs and the WCA and WA values of coating materials can be modeled using a power function with equations y = 37.28x^0.18^ and y = 166.06x^−1.24^, yielding correlation coefficients of 0.701 and 0.986, respectively. The relationship between the N release periods of CRFs and the EB values of films can be modeled using a linear equation (y = −19.42 + 2.57x) with a correlation coefficient of 0.737; furthermore, no significant correlation was observed between the N release periods and TS values for the films tested in this study. Therefore, we selected EB as the characteristic parameter representing the mechanical properties of these films.

Based on the functional equations presented in Figure 4, the values of the material parameters required in order to obtain films with N release periods of either 30 or 90 days can be calculated and are shown in Table 3. The results indicate that for a CRF with an N release period of 30 days, coating materials should have a WA value of <2.4%, WCA value of >68.8°, and EB value of >57.7%. Specifically, when the WA is less than 0.6%, the WCA is greater than 83.8°, and the EB is increased to 211.9%, it becomes possible to produce a CRF with excellent controlled-release properties. 

The results of this study demonstrate the theoretical feasibility of developing a CRF that aligns with the required nutrient release period, provided that the physical properties of the films meet the calculated values. Considering the potential limitations encountered in calculating this value due to objective factors (such as the limited availability of CRFs for factory production), we propose a recommended value (Table 3) based on the fitting results and actual development scenarios for CRFs. Consequently, we conclude that maintaining a WA below 3.0%, a WCA above 60.0°, and an EB above 30.0% is essential for ensuring the film’s compliance with controlled-release requirements.

The results presented in Table 2 and Table 3, as well as Figure 4, demonstrate the significant influence of the mechanical properties of coating materials on their corresponding N release period. In this study, both the PCL1 and PTMG films exhibited WCAs close to the calculated value required for producing a CRF with a N release period of 60 days. However, their actual N release periods were approximately 30 days due to their lower EBs compared to those calculated for a CRF with a N release period of 30 days. Moreover, the PTMG film showed an extended N release period compared to PCL1 due to its superior mechanical properties. Despite having lower hydrophilicity than the PCL1 and PTMG films, the SAL1 film demonstrated higher N release periods in SALCF1 formulations, exceeding 30 days due to its exceptional mechanical properties.

The results of this study demonstrate the theoretical feasibility of developing a CRF that aligns with the required nutrient release period, provided that the physical properties of the films meet the calculated value. Considering the potential limitations in calculating this value due to objective factors (such as the limited availability of CRFs for factory production), we propose a recommended value (Table 3) based on the fitting results and actual development scenarios for CRFs. Consequently, we conclude that maintaining a WA below 3.0%, a WCA above 60.0°, and an EB above 30.0% is essential for ensuring film compliance with controlled-release requirements.

### 3.5. The Controlled-Release Performance and Physical Properties of MSCFs

In order to validate the influence of hydrophobicity and mechanical properties on the controlled-release performance of CRFs, we modified the S material, which exhibited deficiencies in both its hydrophobicity and mechanical properties. Consequently, corresponding CRFs were produced. The properties of both the modified and unmodified films are presented in Table 4, while Figure 5 illustrates the cumulative release curves for N.

The release curve of the SCF in water, as depicted in Figure 5, exhibited a logarithmic trend with *η_t_*_1_ up to 15%. However, after 4 days at a releasing rate of 25%, it became challenging for the SCF to continue releasing nutrients, indicating an uncontrolled performance. In terms of material properties, the SCF film demonstrated high hydrophobicity (with a WCA of 92.1° and WA of 0.1%), but low TS and EB values, which were nearly negligible. When cracks appeared on the shell, a rapid infiltration of water occurred, leading to rupturing and the complete release of nutrients from individual particles. Conversely, when the shell remained intact, no infiltration occurred, and individual particles remained in perfect conditions without nutrient release. By combining S with PCL1-based PU, a soft S/PU hybrid film was obtained with increased WA values approaching the recommended levels while reducing the WCA values accordingly. However, the incorporation of PU resulted in an increase in the EB from 0 to either 10.4 or 48.3%, depending on the amount added; correspondingly, the TS also increased from 0 to either 1.2 or 2.8 MPa, respectively, thus significantly improving film flexibility upon the addition of PU content. When maintaining a ratio of S to PU at approximately equal proportions (i.e., at a ratio of approximately 1:1), both EB and TS reached their recommended values while achieving an extended N release period for MSCF1 that reached up to 40 days, thereby demonstrating good controllability for the S/PU film system, as shown in Figure 5. 

## 4. Discussion

According to the assumptions of previously reported controlled-release models of CRFs [16,19], it is essential for the film to possess both a homogeneous structure and stable performance. It is widely recognized that the film’s structure is determined by the production process, while its stability relies on the molecular properties, hydrophobicity, and mechanical stability of the coating materials. Our research has confirmed that coating materials exhibiting an excellent controlled-release performance should demonstrate adequate hydrophobicity [47] and mechanical properties simultaneously. 

Considering the design of controlled-release films, ideal coating materials should fulfill three essential conditions to meet the crop nutrient release requirements: (a) insolubility in water; (b) excellent hydrophobicity; and (c) adequate mechanical stability. Based on the correlation between the film properties and nutrient release performance evaluated in this study, Figure 6 illustrates three types of controlled-release films immersed in water.

Type I: stable. When the WA is <3%, the WCA is >70°, and the EB is >30%, the film exhibits enhanced hydrophobicity and mechanical properties, simultaneously meeting conditions a, b, and c. Consequently, water permeability through the film becomes significantly hindered [18], while its pores serve as channels for nutrient release [48]. The rate of nutrient release is primarily determined by larger pores [16].

Type II: rupture. When the WA is <1%, the WCA is >80°, and the EB is ≈0, the film exhibits hydrophobic and nonabsorbent properties, thereby satisfying conditions a and b; however, it demonstrates inadequate mechanical characteristics. In its intact state, the film effectively segregates water from the nutrient, impeding the release of the core nutrient. Conversely, when the film is defective, it becomes susceptible to cracking and facilitates the rapid dissolution and release of the nutrient [49].

Type III: swell. When the WA is <100%, the WCA is <60°, and the EB is >30%, the film exhibits hydrophilicity and undergoes swelling in water [50], leading to a compromised barrier capacity within a short period, thereby demonstrating inadequate performance in terms of controlled release [51].

Coating materials matching type I primarily consist of organic synthetic polymers, such as PE and PU films. Meanwhile, certain materials matching type II are predominantly composed of inorganic coated substances, like S film. On the other hand, materials fitting into type III mainly encompass highly water-soluble polymers such as PVA film. In terms of practical applications, SCF is exclusively employed as a slow-release fertilizer. Although water-soluble polymers [51,52] and natural biomaterials [53] are environmentally friendly options, they possess high WA values and fail to meet the actual demands of agricultural production when used for coating CRFs.

At present, the predominant commercially available CRFs are primarily coated with PE or PU polymers. As agricultural chemical production involves the extensive use of polymer-coated CRFs in farmland applications, the quantity involved is substantial, resulting in higher prices. For instance, MDI costs approximately CNY 20,000/ton and polyol costs around CNY 15,000/ton. The addition of PU during CRF production with a coating rate of 4% incurs a cost of CNY 1400/ton. To address this issue, researchers have been exploring various approaches to reduce the cost of polymer-coated fertilizers by minimizing coating rates and optimizing film structures [54]. In comparison to organic materials, inorganic materials offer unparalleled price advantages. S, for example, is priced at only about CNY 2000/ton, and adding S during CRF production with a coating rate of 10% amounts to approximately CNY 200/ton—one-tenth of the price of PU. Therefore, we attempted to establish recommended values for film materials’ properties to guide modifications aimed at rectifying deficiencies while preserving the favorable ecological characteristics inherent in inorganic materials. The results pertaining to the physical properties of modified-S-based materials demonstrate that aligning their material properties with the recommended values proposed herein can enhance the release performance of CRFs. Furthermore, our experiment substantiated that modifying coating materials to achieve these recommended values serves as a fundamental guarantee for producing CRFs exhibiting excellent controlled-release performance.

The modification of vegetable-oil-based PUCFs using modified bentonite resulted in a reduction in WA from 1.3 to 0.3%, an increase in WCA from 95.4 to 97.5°, and an improvement in EB from 12.3 to 20.7%. Consequently, the N release period increased from 23 to 74 days [55]. It is evident that the modified vegetable-oil-based PUCFs meet the recommended values for WA and WCA suitable for a N release period of 90 days; however, the EB falls short of our recommended value, resulting in a shorter N release period than desired (less than 90 days). If the EB can be enhanced to reach 100%, it would extend the N release period beyond 90 days. Therefore, obtaining these recommended values provides clear guidance for selecting and modifying appropriate inorganic materials or biodegradables, which serves as crucial reference information for future research work or industrialization efforts related to CRFs.

## 5. Conclusions

This study has investigated the relationship between the hydrophobicity and mechanical properties of coating materials and the N release periods of commercially available CRFs. The results show significant differences in the N release characteristics of CRFs coated with different materials. Longer N release periods are associated with lower WA, larger WCA, and higher EB values in the corresponding coating material. The correlation coefficients between the N release period and the WA, WCA, and EB were 0.986, 0.701, and 0.737, respectively. Based on the simulated equations, the physical properties for a coating material that can meet the release criteria include a WA less than 2.4%, a WCA greater than 68.8°, and an EB greater than 57.7%. For CRFs demonstrating a N release period of up to 30 days, the recommended values for the coating materials were WA <3.0%, WCA >60.0°, and EB >30.0%. Improving the hydrophobicity and mechanical properties of coating materials according to these recommendations could significantly enhance their controlled-release performance.

## Figures and Tables

**Figure 1 polymers-15-04687-f001:**
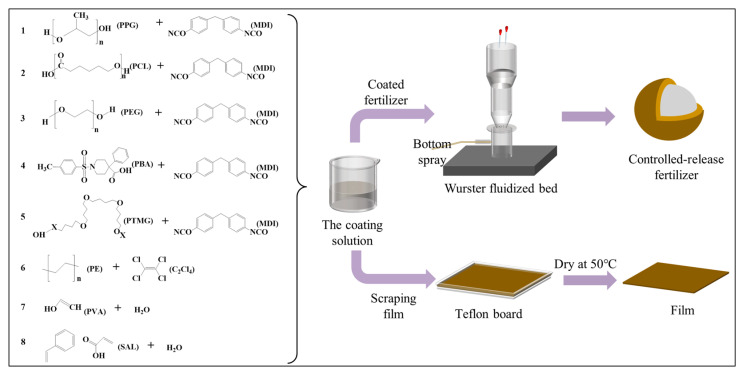
Schematic diagram of the preparation process of CRF and film. Note: NO. 1–5 are the formulations of five PU coating solutions (PPG, PCL, PEG, PBA and PTMG); NO. 6 is the formulations of PE coating solutions; NO. 7 is the formulations of PVA coating solutions; NO. 8 is the formulations of SAL coating solutions.

**Figure 2 polymers-15-04687-f002:**
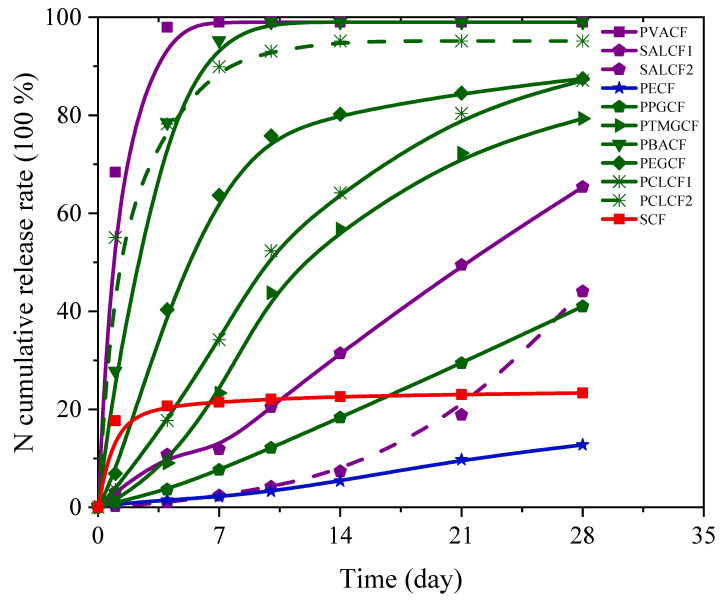
N cumulative release curves of different CRFs.

**Figure 3 polymers-15-04687-f003:**
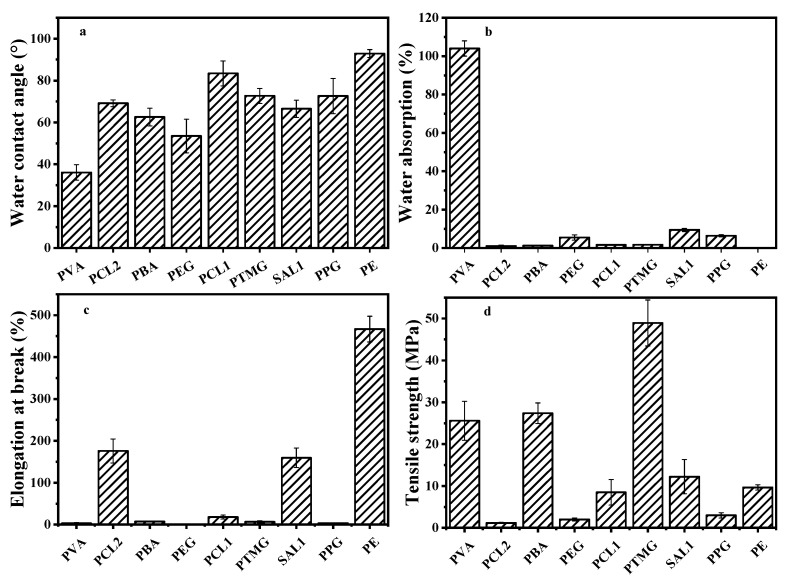
The hydrophobicity and mechanical properties of different coating materials. The hydrophobic properties for different coating materials are presented in (**a**) (WCA), (**b**) (WA), and the mechanical properties for different coating materials are presented in (**c**) (EB), (**d**) (TS).

**Figure 4 polymers-15-04687-f004:**
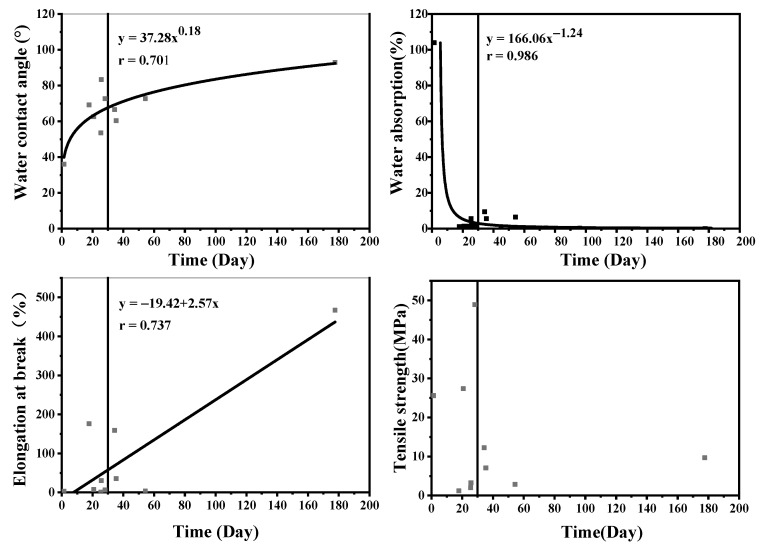
Correlation analyses of the N release periods of CRFs with the physical properties of their coating materials. Note: the squares in the figure are the coating materials with different release periods, and the axis of X = 30 is represent the release period of 30 days.

**Figure 5 polymers-15-04687-f005:**
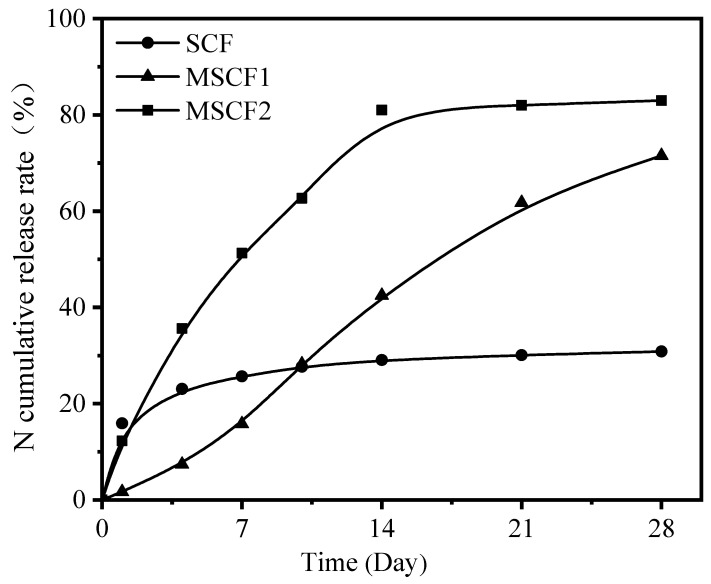
N cumulative release curves of unmodified and modified CRFs.

**Figure 6 polymers-15-04687-f006:**
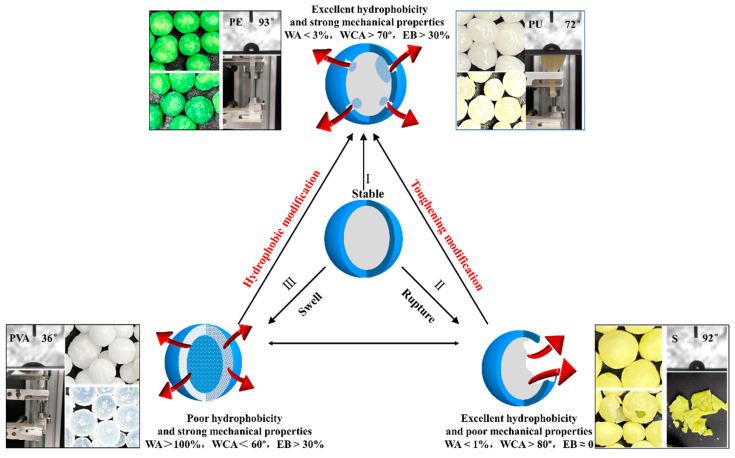
Schematic of the nutrient release mechanism of CRFs as influenced by their coating’s material properties.

**Table 1 polymers-15-04687-t001:** Formula scheme of CRFs.

NO.	Treatment	Quantity of Coating Materials	Urea (kg)	Coating Rate (%)	Temperature (°C)
1	PVACF	1000 g PVA solution (5% *w*/*w*)	1	5	80
2	SALCF1	267 g SAL1 solution (30% *w*/*w*)	1	8	45
3	SALCF2	267 g SAL2 solution (30% *w*/*w*)	1	8	45
4	PECF	625 g PE solution (8% *w*/*w*)	1	5	85
5	PPGCF	19.96 g PPG, 20.04 g MDI	1	4	70
6	PTMGCF	19.88 g PTMG250, 20.21 MDI	1	4	70
7	PBACF	30.8 g PBA 1000, 9.2 g MDI	1	4	70
8	PEGCF	18 g PEG200, 23.8 g MDI	1	4	70
9	PCLCF1	25.76 g PCL3050, 16.6 g MDI	1	4	70
10	PCLCF2	32 g PCL2105, 8 g MDI	1	4	70

Note: The coating rate is the weight ratio of the coating materials (without solvent) to the urea.

**Table 2 polymers-15-04687-t002:** N release parameters of CRFs.

Coating Materials	Treatment	Time							*η* * _*△*t_ *	*T*
		1 d	4 d	7 d	10 d	14 d	21 d	28 d		
PVA	PVACF	68.4	98.1	99.0	99.0	99.0	99.0	99.0	--	1.5
SAL	SALCF1	0.2	1.0	2.4	4.2	7.4	18.9	44.1	1.6	50
	SALCF2	3.1	10.8	11.9	20.5	31.4	49.5	65.4	2.3	34
PE	PECF	0.6	1.5	2.2	3.3	5.4	9.7	12.8	0.4	178
PU	PPGCF	1.0	3.6	7.6	12.1	18.3	29.4	41.0	1.5	54
	PTMGCF	1.2	9.0	23.3	43.8	56.8	72.3	79.3	2.9	28
	PBACF	27.8	78.5	95.3	99.0	99.0	99.0	99.0	16.9	4
	PEGCF	6.9	40.3	63.7	75.8	80.2	84.5	87.5	3.0	25
	PCLCF1	3.7	17.8	34.2	52.3	64.2	80.4	87.1	3.1	26
	PCLCF2	55.1	78.2	89.9	93.1	95.2	95.2	95.2	7.7	4
S	SCF	17.7	20.7	21.5	22.1	22.6	23.0	23.4	--	--

Note: *η_*∆*t_* is the average release rate and *T* is the N release period.

**Table 3 polymers-15-04687-t003:** Parameters of the hydrophobicity and mechanical properties of coating materials.

	Kinetic Equation	r	Release Period (Day)			
			30		90	
			y_1_	y_2_	y_1_	y_2_
WA (%)	y = 166.06x^−1.24^	0.986	2.4	3.0	0.6	1.0
WCA (°)	y = 37.28x^0.18^	0.701	68.8	60.0	83.8	80.0
EB (%)	y = −19.42 + 2.57x	0.737	57.7	30.0	211.9	100.0

Note: the calculated values were obtained by substituing the N release periods into the kinetic equation; x is the N release period and y is the parameter. y_1_ is the calculated value and y_2_ is the recommended value.

**Table 4 polymers-15-04687-t004:** The parameters of physical properties for different modified and unmodified films and the N release period of CRFs.

Treatment	WA (%)	WCA (°)	EB (%)	TS (MPa)	Release Period (Day)
SCF	0.1	92.1	--	--	--
MSCF1	2.4	63.3	48.3	2.8	40
MSCF2	2.2	75.9	10.4	1.2	13

## Data Availability

The data presented in this study are available in the article.

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
