# Peer review of "Recommended Values for the Hydrophobicity and Mechanical Properties of Coating Materials Usable for Preparing Controlled-Release Fertilizers"

_polymers, 2023, doi:10.3390/polym15244687_

Round 1
Reviewer 1 Report (Previous Reviewer 1)
Comments and Suggestions for Authors
The revised version of the manuscript has significant improvements. First of all the authors discussed the cost efficiency aspects, secondly the entire text was improved.
However so small changes are still required:
1) Fig5 is not readable, while this is the most important graph
2) The authors indicate dhigh cost of the polymer and cost barrier for PU CRF. But please add your vision of the solution for such problem. E.g. decrease of the polymer coating thickness, changing of polymer nature, etc.
Comments on the Quality of English LanguageThe quality of English is OK
Author Response
回复 1:Thanks for your feedback, I have redrawn Figure 5 to make it clearer.
回复 2:Thank you for your advise, We have added it, please see Line 490-492 of Rev. MS.

Reviewer 2 Report (Previous Reviewer 3)
Comments and Suggestions for Authors
The article has been proofread. The authors introduced significant changes in several areas. However, there are still some aspects that need improvement:
Materials and Methods: there is no information on estimating analysis/measurement errors
Results: only the results should be presented - the authors should decide whether they present them in the form of a table, graph or text, duplication of information is not necessary. the authors should move the content relating to the discussion of the obtained results to chapter 4. Discussion or chapters 3 and 4 should be combined: Results and Discussion - then there would be no need to move 4.Discussion
Discussion: The discussion of the results is very poor - this is probably due to the fact that part of the discussion is presented in the Results chapter. However, the authors should refer to other results presented in this area. A reference to only few articles and an analysis in this context of only few selected results obtained from the analyzes performed is not enough. The authors have interesting results, but the results should be analyzed by comparing them to other results and trying to determine which information is relevant and in what context.
Author Response
Respones 1: Thank you for your positive comments and your valuable suggestions on the manuscript, and we would like to express our sincere thanks for your advice. We have added it, please see Line 255 and 270-275 of Rev. MS.
Respones 2: Thanks for your suggestion. We have moved part of the content relating to the discussion of the obtained results to Line 498-500 of Rev. MS. But some parts in chapter 3.4 are to get the recommended value to match the results of modified CRFs in chapter 3.5. So we think it’s much better for keeping these parts.
Respones 3: Thanks, we have improved the discussion as your suggestion. In the discussion, we emphasized the thresholds that the performances of coating materials which includes three parts of water absorption, water contact angle and elongation at break must reach, and neither part of these three parts alone can be established for the recommendation. We have also added reference to further elucidate the issue[54], Which can analyzed that comprehensive regulation of material properties can help the CRFs get better controlled release performances in the introduction. Therefore, in the discussion, our main purpose is to forecast the importance of the recommended value of material properties for the cost reduction and efficiency increase in the industry of CRFs.
Reference
- H. Zhao; Y.Q. Wang; L.X. Liu; L.X. Liu; M. Chen; C.Q. Zhang; Q.M. Lu. Green coatings from renewable modified bentonite and vegetable oil based polyurethane for slow release fertilizers. Polym. Comp. 2017, 39, 4355-4363. https://doi.org/10.1002/pc.24519.

This manuscript is a resubmission of an earlier submission. The following is a list of the peer review reports and author responses from that submission.
Round 1
Reviewer 1 Report
Comments and Suggestions for Authors
This manuscript presents very important practical results dealing with the CRFs optimization via wettability adjustments. All results are very important and will attract significant attention from engineers ad scientists. However, some issues must be addressed:
1) The correlation between WCA and WA of coating materials presented in Figure 4 must be revised. Indeed, the hydrophobic results are covered with only one point. Thus your fit is quite questionable, as the range of WCAs is not well adjusted.
2) Please provide some cost estimations for the proposed CRFs.
3) Highlight key challenges for cost reduction
Comments on the Quality of English LanguageEnglish is fine
Reviewer 2 Report
Comments and Suggestions for Authors
he current version of this manuscript is not apt for the revision. The authors must work in clarifying this version to enable its reading. First, when one reads the introduction it is hard to know what are they going to do next. Some materials are missing in the materials section, some acronyms are not defined, the references are not well cited (the number does not correspond with the reference list and some DOI are repeated), the coating technique is so confused that I stopped reading.
Comments on the Quality of English LanguageExtensive editing is required
Reviewer 3 Report
Comments and Suggestions for Authors
The introduction should be redrafted: the authors describe the issues related to the modification very extensively, but refer to the topic to a small extent.
There is no research hypothesis. The authors in a few sentences wrote what they did. But they did not present a hypothesis for the study (in the context of the proposed title of the manuscript).
Section 2.1 should have a different title as the devices are also shown.
The methodology does not include information on validation, while in the "Results" section, the authors indicated that they performed validation. However, it is not known what this validation was about, what the modification consisted of and how it was carried out. In addition, validation is based on reference materials, not modified ones.
How is the WA of films marked? (the form of the sentence indicates that a given method could be used, but it is not known whether it was chosen and applied).
The results should be presented using one form - presentation in tables and graphs as well as in the text of the same information is redundant. The summary, presented on page 8, should be removed - the section 3 is to show the results, the summary is a separate chapter. The same applies to the subsequent summaries contained in the following subsections.
The "Results" section should show the results and their interpretation should be in a separate chapter (e.g. Discussion of results). Or change the title of the chapter to "Results and Discussion".
In order to be able to conclude that only one form/type of material should dominate the market, it would be necessary to analyze all those available on the market (and not only selected ones) and test them all. The authors, on the other hand, selected basic materials and evaluated these materials in terms of various (selected) parameters. In addition, the studies did not take into account the parameters affecting the release process. Therefore, such wordings cannot be made.
Moreover, it is not known how the indicated works relate to the title of the article. The discussion of the results is very poor. The authors presented three types of CRF, however, they did not refer to other divisions published in various literature items.
The authors recommend values, however, it is not known what assumptions were made, on what basis, whether the recommended values should apply to all materials on the market or only to those selected by the authors, and how it relates to the fact that the release is affected by temperature, pH environment, etc. The authors do not take this into account at all.
The methodology adopted by the authors is not adequate to the assumptions/title of the article. To propose something, the parameters relevant to the release process must be taken into account, but the authors did not do.
